# Sweeping Heterogeneity with Smart MoPs: Mixture of Prompts for LLM Task Adaptation

## Abstract

Large Language Models (LLMs) have the ability to solve a variety of tasks, such as text summarization and mathematical questions, just out of the box, but they are often trained with a single task in mind. Due to high computational costs, the current trend is to use prompt instruction tuning to better adjust monolithic, pretrained LLMs for new –but often individual– downstream tasks. Thus, how one would expand prompt tuning to handle –concomitantly– heterogeneous tasks and data distributions is a widely open question. To address this gap, we suggest the use of *Mixture of Prompts*, or MoPs, associated with smart gating functionality: the latter –whose design is one of the contributions of this paper– can identify relevant skills embedded in different groups of prompts and dynamically assign combined experts (i.e., collection of prompts), based on the target task. Additionally, MoPs are empirically agnostic to any model compression technique applied –for efficiency reasons– as well as instruction data source and task composition. In practice, MoPs can simultaneously mitigate prompt training "interference" in multi-task, multi-source scenarios (e.g., task and data heterogeneity across sources), as well as possible implications from model approximations. As a highlight, MoPs manage to decrease final perplexity from $\sim 20\%$ up to $\sim 70\%$, as compared to baselines, in the federated scenario, and from $\sim 3\%$ up to $\sim 30\%$ in the centralized scenario.

## 1 Introduction

**Background.** Recent advances in large language models (LLMs) demonstrate that they are powerful general-purpose models; for some, it is believed that, through LLMs, we are getting closer to the holy grail of task-agnostic Artificial General Intelligence (AGI) (Brown et al., 2020; Bommasani et al., 2021; Bubeck et al., 2023). A factor towards such a belief is the ability of LLMs to solve drastically different instructed tasks out of the box –often known as *emergent abilities* (Wei et al., 2022a), which in turn are also under criticism (Schaeffer et al., 2023)– ranging from text summarization (Goyal et al., 2022; Liu & Lapata, 2019; Bubeck et al., 2023) to solving mathematical questions (Shi et al., 2022; Lee et al., 2023; Bubeck et al., 2023).

Yet, despite this success, recent studies put LLMs' performance under the spotlight on a broad set of tasks, hinting that their task-agnostic ability might be brittle. Summarizing these results, one might observe that $i$) small changes on handcrafted task prompts (Zhao et al., 2021; Holtzman et al., 2021) and/or $ii$) changes in the model size and function family class (Ouyang et al., 2022b; Wei et al., 2022b) and/or $iii$) the use of model compression techniques to save computation costs (Xu et al., 2023), all result in –often non-negligible– performance variability, if not deterioration. As such, there is an inevitable trade-off between accuracy and efficiency, resulting in a decrease in the overall performance of LLMs.

The ML/AI community has responded to these challenges. For instance, (soft) prompt instruction tuning –based on downstream task data– is proposed to better fine-tune deployed models, in order to adjust to –often individual– downstream tasks (Ouyang et al., 2022a; Kenton et al., 2021; Bender et al., 2021; Tamkin et al., 2021). Similar –and relatively concurrent– attempts created the term *parameter-efficient fine-tuning* (PEFT) methods (Houlsby et al., 2019; Ding et al., 2023), including adapter tuning (Houlsby et al., 2019; Hu

et al., 2023), prefix tuning (Li & Liang, 2021), prompt tuning (Lester et al., 2021), low-rank adaptation (LoRA) (Hu et al., 2021), and compression aware prompts (Xu et al., 2023), among others.

**A gap that persists.** Yet, the scenarios considered as part of the above studies do not correspond to some practical scenarios found in reality. For instance, consider the following LLM instruction tuning scenario: Company `XYZ` intends to develop a general purpose office assistant application that solves different types of assistant tasks. For these targeted tasks, company `XYZ` uses human "labelers" to generate demonstration data of related tasks; these are stored in a central server. At the same time, company `XYZ` agrees with clients to locally utilize their own local demonstration data (i.e., previous human assistant work record). Overall, company `XYZ` desires to decrease both training and inference cost of the final model, *by aiming in the generation of specialized "experts" that can be used on-the-fly and just-in-time for most incoming clients, without necessarily requiring further fine-tuning.*[1]

What company `XYZ` is facing is the following challenge: *Can existing prompt-tuning strategies utilize all the available data from both central server and local clients to construct specialized experts –instead of randomized ones– while maintaining desirable computation/communication costs?* Such scenarios suggest a *multi-source, multi-task prompt tuning* approach, which includes both centralized training and federated learning scenarios as special cases. The emphasis though is in the training of specialized prompts that operate in a modular way such that, when combined together, they tackle tasks in a just-in-time manner.

While multi-task learning and multi-source learning in LLMs has been considered in the past (Radford et al., 2021; Reed et al., 2022; Huang et al., 2023; Bubeck et al., 2023), to the best of our knowledge, there is limited work on PEFT methods that satisfy the above desiderata. From the federated learning perspective, (Babakniya et al., 2023; Chen et al., 2023) considers the federated version of LoRA (Hu et al., 2021); (Zhang et al., 2023) considers the federated version of adapters; while (Jiang et al., 2023) suggests on-going pretraining of the full-model for better domain adaptation, based on the findings in (Gururangan et al., 2020). Yet, to the best of our understanding, *these works focus mostly on the periodic aggregation and averaging of the PEFT-based parameters*, without targeting necessarily on specialized experts (i.e., prompts) that –when combined– outperform on just-in-time tasks, based on compressed models. Other concurrent work on multiple prompts, as in (Si et al., 2023; Asai et al., 2022), assumes a prior knowledge of skills/tasks and uses hand-designed "expert" prompts. The latter works also do not consider multi-source data heterogeneity, while (Si et al., 2023) uses a limited capacity random forest as a gating function.

**Overview of our approach and contributions.** Inspired by work on mixture of experts, we propose to use *Mixture of Prompts* (or MoPs) in *multi-source, multi-task prompt instruction tuning*, in order to efficiently leverage all available data from both the central server and local clients. Our hypothesis is that key obstacle in such settings is the appearance of implicit "interference" during training; see the sections that follow. In this work, the use of MoPs is guided by a novel gating functionality that can identify relevant skills embedded in different groups of prompts ("experts" in this work), based on the data domain of the current input, and dynamically selecting the combination of relevant prompts. This is in stark contrast with existing work on mixtures of prompts, where one naively aggregates the updated prompts that have been simultaneously trained on different tasks and/or diverse data distributions. Our contributions are threefold:

- **Tackling task/data heterogeneity.** We design MoPs with the property of being agnostic to the training instruction data source. MoPs could utilize either solely centralized data, collected by human "labelers", or heterogeneous local data (e.g., stored on edge devices), or a combination of those, while being agnostic about the composition of instruction data.

- **Model compression resiliency.** Via experiments, we have observed an emerging ability of MoPs: they work *out of the box*, regardless of any reasonable model compression ratio or technique (i.e. pruning, quantization). MoPs consistently outperform existing baselines across various metrics and datasets, demonstrating its effectiveness and robustness.

---

[1]The definition of an "expert" here will be apparent later on in the text; this should not be necessarily assumed as MLP experts in sparse mixture of experts (Puigcerver et al., 2023).

- **Empirical performance.** As a highlight of our results, MoPs manage to decrease final perplexity from $\sim 20\%$ up to $\sim 70\%$, as compared to baselines, in the federated scenario, and from $\sim 3\%$ up to $\sim 30\%$ in the centralized scenario. Our gains in the federated setup further support our hypothesis that our gating function overcomes data heterogeneity under highly skewed distributions, reducing the model drift problem.

## 2 BACKGROUND AND RELATED WORK

**LLMs and Decoder-only Transformers.** The backbone of LLMs are decoder only transformers (Vaswani et al., 2017; Liu et al., 2018). A LLM takes as input a question (along with question context) and performs next word prediction to generate answers/response for the question.

$$A^h = \texttt{Softmax}\left(M\left(W_q^{h}X^{\ell\top}(W_k^{h}X^{\ell})\right)\right) \in \mathbb{R}^{n \times K};$$

$$\widehat{V}^h = A^h\left(W_v^{h}\texttt{Concat}(P^{\ell}, X^{\ell})\right) \in \mathbb{R}^{d_h \times n};$$

$$O = W_o\texttt{Concat}\left(\widehat{V}^0, \widehat{V}^1, \ldots, \widehat{V}^H\right) \in \mathbb{R}^{d_t \times n};$$

$$X^{\ell+1} = W_{\text{ff2}}(\texttt{Relu}(W_{\text{ff1}}O)).$$

The forward pass of the $\ell$-th layer is shown in the wrapped equations of this paragraph. Let $d_h$ denote the dimension of the attention head, $d_t$ the dimension of the input token embedding, $d$ the width of the feedforward layer, $H$ the number of attention heads and $n$ the input sequence length. Finally, $M$ is the decoder attention mask.

**LLMs with Trainable Prompts:** Following (Ouyang et al., 2022a; Kenton et al., 2021; Bender et al., 2021; Tamkin et al., 2021), we consider trainable prompts to perform efficient instruction tuning on LLMs. Using similar notation and additional $K$ trainable prompts $P^0 \in \mathbb{R}^{d_t \times K}$, the forward pass of the $\ell$-th module can be formulated as below:

$$B = \texttt{Concat}(P^{\ell}, X^{\ell}) \in \mathbb{R}^{d_t \times (n+K)}$$

$$C = \texttt{Concat}(P^{\ell}, X^{\ell}) \in \mathbb{R}^{d_t \times (n+K)}$$

$$A^h = \texttt{Softmax}\left(M'\left(W_q^{h}B^{\top}(W_k^{h}C)\right)\right) \in \mathbb{R}^{(n+K) \times (n+K)};$$

$$\widehat{V}^h = A^h\left(W_v^{h}\texttt{Concat}(P^{\ell}, X^{\ell})\right) \in \mathbb{R}^{d_h \times (n+K)};$$

$$O = W_o\texttt{Concat}\left(\widehat{V}^0, \widehat{V}^1, \ldots, \widehat{V}^H\right) \in \mathbb{R}^{d_t \times (n+K)};$$

$$\texttt{Concat}(P^{\ell+1}, X^{\ell+1}) = W_{\text{ff2}}(\texttt{Relu}(W_{\text{ff1}}O)),$$

where $W_q^h$, $W_k^h$, $W_v^h$, $W_{\text{ff1}}$, $W_{\text{ff2}}$ are all **frozen**, while $P^0$ is the only trainable set of parameters during training. After first layer, we treat $P^{\ell}$ as normal tokens embedding through LLM layers. $M'$ is the modified decoder attention mask where all prompts are never masked out for all input tokens. Here, $\texttt{Concat}(B, C)$ –with $B$ and $C$ of appropriate dimensions– concatenates the two matrices columnwise. We omit skip connections and layer normalization modules to simplify notations.

**Injection of prompts.** Inspired by experiments in (Li & Liang, 2021), we further propose to inject trainable prompts in middle layers. An illustrative example can be seen in Figure 1. The benefits of this design are twofold. First, it reduces the computational cost of training by reducing the number of layers that need to be backpropagated. Second, it allows for greater flexibility in the design of the model architecture, as the prompts can be placed in any layer, rather than just the first layer.

**Prompt-tuning in Federated Learning:** Recent approaches adapt FedAvg (McMahan et al., 2017) to prompts tuning (Zhao et al., 2023; Babakniya et al., 2023). During the local training phase, each client will optimize the local copy of prompts. During synchronization, all updated copies of prompts are averaged on the server for the next round of training. *This is in stark contrast with this work: while the idea of mixing prompts is not new, we are focusing on learning relevant skills as expressed via selected subsets of prompts, based on the data domain of the current input and dynamically selecting the combination of relevant prompts to solve current and new tasks.*

## 3 MIXTURE OF PROMPTS (MoPs) WITH A SMART GATING FUNCTION

**Our hypotheses in a nutshell.** Current prompt tuning approaches (both centralized and federated) might not operate to their full potential, especially when facing task heterogeneity

(i.e., when training involves multiple tasks simultaneously), data heterogeneity (i.e., when training with imbalanced data, e.g., across distributed clients), and when approximate (e.g., compressed) models are in use to further reduce computation costs.

Our hypothesis is that training prompts to handle *universally* multi-source multi-task scenarios might result in *prompt interference* across tasks and across sources. More specifically, one way that *prompt interference* can be decomposed is as follows:

- In centralized training, prompts might converge to poor-performing parameter configurations, when heterogeneous tasks are considered, due to *conflicting training signals from different tasks*. This case is especially challenging when the tasks are distinctly diverse.
- In privacy-preserving scenarios, such as federated learning, *heterogeneous data distributions* add more training interference across clients. The model can be biased towards the tasks with more data, losing its capability for generalization.
- For efficiency reasons, compressed LLMs are now widely used for both centralized and federated learning scenarios. Such model approximations could impose implicit prompt training interference, since trainable prompts are responsible for *both recovering model capacity loss –due to compression– and model adaptation for downstream tasks.*

**Algorithm desiderata.** Given the above, the designed methodology should: *i*) be able to learn from scratch a diverse set of "skills", that will be embedded in different prompts to avoid interference, or help to recover such "supressed" skills due to model compression; *ii*) dynamically select and combine the prompts with relevant skills for any incoming input data; the latter is in contrast to existing methods that often use all prompts for all subtasks during training and testing.

## 3.1 Mixture of Experts (MoPs) Design

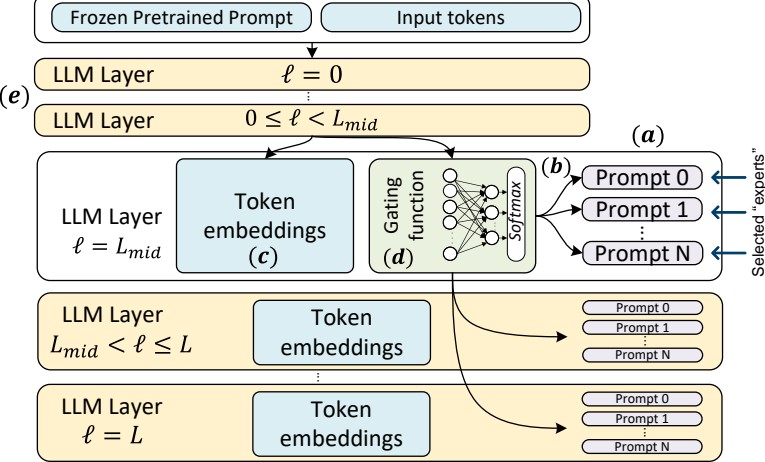

Figure 1: Mixture of Prompts with a Smart Gating Function on Compressed LLMs overview.

**Prompts as experts.** To embed different skills across subtasks, we utilize multiple trainable prompts as experts, each being a collection of prompts specializing on different skills; see Figure 1(a). These prompts are then selected by the gating function (see Figure 1(b), and as described below), depending on the current input; see Figure 1(c). This allows us to use different combinations of skills, embedded in prompts, for different input tasks, resulting in a more accurate handling of incoming tasks. Per iteration, a subset of prompts is selected to be updated, which avoids the training interference between prompts. Due to consideration of further reduce the training cost, we inject the prompts in the middle layer as discussed above.

**The gating function.** To dynamically select expert prompts based on current input question/task, we design a gating function that embeds the current question. In order to avoid paying extra computation/memory cost by using another independent embedding network as in common Mixture of Expert practice, our gating function directly uses first half of the given model ($0 \leq \ell \leq L_{\mathrm{mid}}$) as the embedding network without any additional

cost; see Figure 1(e). Our gating function utilizes a shallow MLP network with softmax layer to generate expert score for each expert prompt, which is used to scale the attention of these expert prompts in the following layers ($L_{mid} \leq \ell \leq L$); see Figure 1(d). The exact mathematical formulation is shown in Algorithm 1. By using softmax-based expert score, the gating function "forces" later layers to only focus on selected prompts, which in turn scale the updates for each prompts accordingly during back propagation. Finally, our gating function imposes a negligible computation overhead in total.

**Pretraining the gating function.** To improve the initial performance of our gating function, we assume we have unlabeled instruction data (instruction/question only) with domain/task labels on server side. As such data are instruction only, we assume that in both centralized and federated learning case we can collect such data beforehand. [2] We use this data to pretrain the gating function by manually assigning a one-to-one relationship between each prompt group and each data domain/task. This provides a good initialization to the gating function, as it assumes that *i*) each subtask is drastically different and represents one distinct skill, and *ii*) each prompt embeds the corresponding skill. Such an assumption does not need to be totally accurate for the available dataset. As shown in the experiments, such an initialization is good enough: eventually, the gating function, together with trainable prompts, are able to discover a more accurate relationship between subtasks; i.e., which skills are shared or not shared between subtasks.

**Using compressed LLMs for efficient prompt tuning.** Due to training efficiency concerns, compressed LLMs are widely used for downstream instruction tuning in both centralized and federated learning scenarios. We follow this paradigm: our system, depicted in Figure 1, utilizes *aggressively compressed LLMs*. To further reduce the computation costs, we strategically add prompts only to the middle layers of the model, thus avoiding back propagation of the full model during training.

The above are summarized in Algorithm 1. Briefly, given an input question, MoP first embeds the question using the first $L_{\mathrm{mid}}$ layers of a given compressed LLM. We set $L_{\mathrm{mid}} = 10$ for a LLama-7B model with $L = 20$. Such choice of $L_{\mathrm{mid}} = 10$ is to balance two conflicting requirements: (1) we want to inject prompts as late as possible to reduce back propagation cost during training and increase depth/capacity for embedding network of gating function (2) prompts should be injected early to have more capacity in influencing the pretrained LLM network. At layer $L_{\mathrm{mid}}$, we inject $K$ trainable prompts. The gating network uses the embedding from the previous layer to generate experts score for each prompt, based on the input question, which is used to re-scale attention weight from other tokens to those prompts. After layer $L_{\mathrm{mid}}$, it follows the normal LLM forward propagation.

## 4 EXPERIMENTS

In this section, we present experiments conducted to evaluate the performance and effectiveness of our method on a variety of tasks and contrast it with baseline approaches. Since approximate LLMs become increasingly valuable in the foreseeable future, due to the faster training and inference times, as well as the significant reduction in energy consumption, we express our results taking into consideration different pruning ratios.

**Datasets.** We evaluated our method using two datasets: Databricks Dolly 15k (Conover et al., 2023) and Super-Natural Instructions (Mishra et al., 2022). Table 1 outlines the seven task categories into which we divided both datasets. These datasets pose a challenge for our method: MoPs have to learn and select relevant skills from scratch, without any prior knowledge of the complex relationships between the subtasks. For the centralized setup, we split the original 5k samples from each dataset into 90% training and 10% testing sets. We used a batch size of 1 for both training and testing. In the federated scenario, we simulated an uneven distribution of data across 100 clients, resulting in different proportions and sizes of data. The batch size remained at 1. The distribution of data skew across clients is explained in Appendix A.

---

[2]We leave for future study in more strict federated learning scenario where such unlabeled instruction are also federated.

---

**Algorithm 1** Mixture of Prompts (MoPs) with a smart gating function

---

Parameters: $\odot$ denotes row-wise element and we replace old prompts with new prompts $\hat{\mathbf{P}}^{L_{mid}}$ in layer $\ell = L_{mid}$

---

**♠ Before middle layer ♠**

**for** $0 \leq \ell < L_{mid}$ **do**

$\quad \mathbf{A}^h = \texttt{Softmax}\left(\mathbf{M}'(\mathbf{W}_q^h(\texttt{Concat}(\mathbf{P}^\ell, \mathbf{X}^\ell))^\top(\mathbf{W}_k^h \texttt{Concat}(\mathbf{P}^\ell, \mathbf{X}^\ell))\right) \in \mathbb{R}^{(n+K)\times(n+K)}$;

$\quad \hat{\mathbf{V}}^h = \mathbf{A}^h\left(\mathbf{W}_v^h \texttt{Concat}(\mathbf{P}^\ell, \mathbf{X}^\ell)\right) \in \mathbb{R}^{d_h \times (n+K)}$;

$\quad \mathbf{O} = \mathbf{W}_o \texttt{Concat}\left(\hat{\mathbf{V}}^0,\ \hat{\mathbf{V}}^1, \ldots,\ \hat{\mathbf{V}}^H\right) \in \mathbb{R}^{d_t \times (n+K)}$;

$\quad \texttt{Concat}(\mathbf{P}^{\ell+1}, \mathbf{X}^{\ell+1}) = \mathbf{W}_{ff2}(\texttt{Relu}(\mathbf{W}_{ff1}\mathbf{O}))$

**end for**

---

**♠ Middle layer ♠**

**if** $\ell = L_{mid}$, where $i \in Q + C$ means only tokens in question and context **then**

$\quad \mathbf{G} = \texttt{Softmax}(\mathbf{W}_{gff2}(\texttt{Relu}(\mathbf{W}_{gff1}(\texttt{Mean}_{i \in Q+C}(\mathbf{X}_i^\ell))))) \in \mathbb{R}^K$

**end if**

---

**♠ After middle layer ♠**

**for** $L_{mid} \leq \ell < L$ **do**

$\quad \mathbf{A}^h = \texttt{Softmax}\left(\mathbf{M}'(\mathbf{W}_q^h(\texttt{Concat}(\hat{\mathbf{P}}^\ell, \mathbf{X}^\ell))^\top(\mathbf{W}_k^h \texttt{Concat}(\hat{\mathbf{P}}^\ell, \mathbf{X}^\ell)))\right) \in \mathbb{R}^{(n+K)\times(n+K)}$;

$\quad \mathbf{A}^h[:, 0:K] = \mathbf{A}^h[:, 0:K] \odot \mathbf{G} \in \mathbb{R}^{(n+K)\times(n+K)}$

$\quad \hat{\mathbf{V}}^h = \mathbf{A}^h\left(\mathbf{W}_v^h \texttt{Concat}(\hat{\mathbf{P}}^\ell, \mathbf{X}^\ell)\right) \in \mathbb{R}^{d_h \times (n+K)}$;

$\quad \mathbf{O} = \mathbf{W}_o \texttt{Concat}\left(\hat{\mathbf{V}}^0,\ \hat{\mathbf{V}}^1, \ldots,\ \hat{\mathbf{V}}^H\right) \in \mathbb{R}^{d_t \times (n+K)}$;

$\quad \texttt{Concat}(\hat{\mathbf{P}}^{\ell+1}, \mathbf{X}^{\ell+1}) = \mathbf{W}_{ff2}(\texttt{Relu}(\mathbf{W}_{ff1}\mathbf{O}))$

**end for**

---

| Dataset | Dolly-15K Instructions | Super-Natural Instructions |
|---------|------------------------|----------------------------|
| Subtasks | creative writing
closed QA
open QA
summarization
information extraction
classification
brainstorming | quoref-question-generation
drop-question-generation
essential-terms-identifying-essential-words
add-integer-to-list
evaluation-semantic-relation-classification
ljspeech-textmodification
mmmlu-answer-generation-global-facts |
| Total | 5000 samples | 5000 samples |

Table 1: Task categories used per dataset

**Setup.** We utilize SparseGPT(Frantar & Alistarh, 2023) to perform structured/unstructured pruning of the LLama-7B model to create an aggressively compressed LLM. Inspired by (Xu et al., 2023), we assign 10 prompts as a single expert, creating in total 7 experts and ensuring a 1:1 relationship between experts and tasks. As shown later in experiment, such 1:1 relationship between experts and tasks are not hard restriction as gating function learns to group tasks and change assigned experts based on their similarity, often using fewer number of experts. Thus when number of tasks are unknown, we can still use fixed number of experts. Due to recent advance in pruned LLM such as paper Xu et al. (2023), we suggest that in the future pruned LLM models might will also come with pretained prompts to partially recover the pruned model performance loss. In order to show our method can even further recover/improve the performance of pruned model, we also add such pretrained prompts to both our baseline and our model. In our experiments, we trained these prompts from scratch in a preprocessing step over 20 training steps. These pretrained prompts are frozen during training. For our method, we add 70 prompts to the mid layer ($L_{\text{mid}} = 10$) and replace the pretrained prompts in the following layers. The gating function is designed to create the prompt/expert weight for each group.

In the centralized setting, we use total 20000 steps with learning rate 0.001. In the federated setting, we adapt FedAvg such that during each synchronization round, we average the updated prompts from all active clients. We use 100 clients, with 10 active clients per

training round, and set each local training round to 250 training steps. Counting all clients, the total number of training steps is 50000 with learning rate 0.001. (Each active client with around 5000 steps in total and 10 active clients at each time)

**Baselines.** A reasonable baseline is to directly apply prompt training to both centralized and federated training without any gating function. In centralized training, we use method from Xu et al. (2023) as baseline. In federated training, we utilize `FedPrompt` from (Zhao et al., 2023), which adapts FedAvg to prompt training and periodically averaging the updated prompts from all clients. In both cases, to match computation and memory cost with our method during training, we add additional prompts in the mid layer and freeze the given pretrained prompts in the first layer, thus eliminating the need to calculate gradients before the mid layer.

**Centralized training results.** In Tables 2 and 3, we present the results of our method applied to different structured/unstructured pruning ratios in the centralized learning scenario. For unstructured pruning, we use $X\%$ to denote pruned model with $X\%$ weight pruned out. For structured pruning, we follow the notation in (Frantar & Alistarh, 2023) to use $N:M$ to denote pruning N elements out of consecutive M elements in weight matrix. We observe that our method achieves a significant reduction in the final PPL for all cases, with a greater advantage for the highest pruning ratios. This supports our claim that our method helps to alleviate prompt training interference as higher pruning ratio increases the "burden" on prompts to recover the skills from model loss while our method reduces such burden on prompts and gives them more capacity for task adaption. Additionally, we note that the PPL reduction in the centralized case is more pronounced for the unstructured pruning, as expected due to lower degree of sparsity.

| | | Unstructured pruning (ratio) | | |
|---|---|---|---|---|
| Dataset | Methods | 90% | 85% | 75% |
| Dolly-15K | Baseline | 52.65 | 18.16 | 8.25 |
| | MoPs | 40.34 | 15.04 | 7.24 |
| | Gain ± | +12.31 (30%) | +3.12 (20%) | +1.01 (13%) |
| Super-Natural | Baseline | 58.47 | 16.50 | 8.54 |
| | MoPs | 52.86 | 14.59 | 7.80 |
| | Gain ± | +5.61 (11%) | +1.91 (13%) | +0.74 (9%) |

Table 2: Summary of final perplexities reported on unstructured pruning in centralized scenario on Dolly-15 and Super-Natural datasets.

| | | Structured pruning (Type & Ratio) | | | |
|---|---|---|---|---|---|
| Dataset | Methods | 7:8 (87.5%) | 3:4 (75%) | 2:4 (50%) | 4:8 (50%) |
| Dolly-15K | Baseline | 70.14 | 9.06 | 3.67 | 3.76 |
| | MoPs | 54.97 | 8.08 | 3.54 | 3.59 |
| | Gain ± | +15.17 (27%) | +0.98 (12%) | +0.13 (4%) | +0.17 (5%) |
| Super-Natural | Baseline | 67.86 | 10.64 | 6.01 | 5.90 |
| | MoPs | 59.80 | 10.05 | 5.79 | 5.73 |
| | Gain ± | +8.06 (13%) | +0.59 (6%) | +0.22 (4%) | +0.17 (3%) |

Table 3: Summary of final perplexities reported on structured pruning in centralized scenario on Dolly-15 and Super-Natural datasets.

**Gating function analysis on the centralized setup.** We further analyze how our gating function performs the assignment depending on the current task. In Figure 2, we observe that the pretraining step helps the gating function to roughly distinguish between data domains/tasks, by encouraging one-to-one relationship between prompt experts and data domains/tasks. After training is done, instead of one-to-one relationship between prompt experts and data domains/tasks, we can see that our gating function learns to select the same expert group of prompts for similar tasks. This suggests that our gating function has learned to adjust the prompt weight distribution, in order to better capture the

domain/task relationship and specialize the expert assignment. Results on more pruning ratios are included in Appendix B and C.

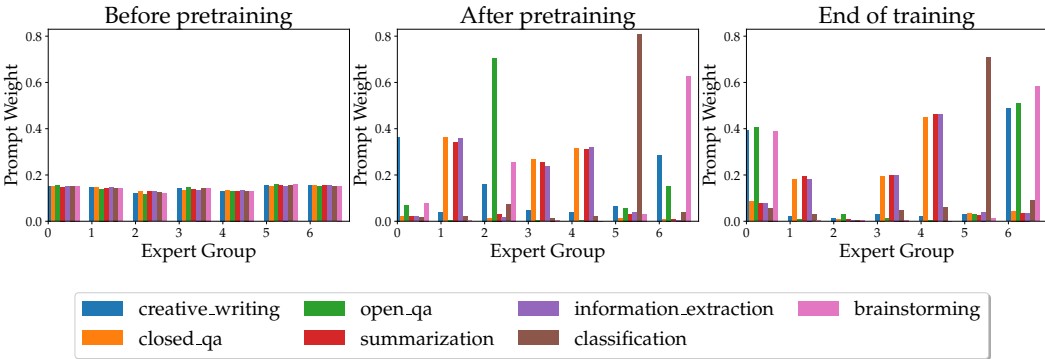

Figure 2: Averaged Prompt weight assigned each prompt group by gating function for test dataset using 85% unstructured pruning Llama-7B in centralized setup.

**Federated training results.** Our method was tested in a federated learning setup, using the same structured/unstructured pruned models with the centralized scenario. The results, presented in Tables 4 and 5, demonstrate that our approach is still superior to the baselines (here, `FedPrompt`) for all the pruning ratios. We included an additional row to highlight the *relative gain* (PPL decrease) of our method in both datasets. When we compare these gains with the ones presented in Tables 2 and 3, it becomes clear that our method in the federated setup, yields superior gains to the baseline in comparison with the previous centralized numbers.

| Dataset | Methods | Unstructured pruning (ratio) | | |
| | | 90% | 85% | 75% |
| --- | --- | --- | --- | --- |
| Dolly-15K | FedPrompt | 98.13 | 28.28 | 11.99 |
| | MoPs | 65.25 | 20.77 | 9.45 |
| | Gain ± | **+32.88 (50%)** | **+7.51 (36%)** | **+2.54 (27%)** |
| Natural Instruction | FedPrompt | 76.17 | 18.64 | 9.14 |
| | MoPs | 66.51 | 16.52 | 7.88 |
| | Gain ± | **+9.66 (15%)** | **+2.12 (13%)** | **+1.26 (16%)** |

Table 4: Summary of final perplexities reported on unstructured pruning in federated scenario,using a pool of 100 available clients, sampling 10 per iteration.

| Dataset | Methods | Structured pruning (Type & Ratio) | | | |
| | | 7:8 (87.5%) | 3:4 (75%) | 2:4 (50%) | 4:8 (50%) |
| --- | --- | --- | --- | --- | --- |
| Dolly-15K | FedPrompt | 143.02 | 17.20 | 5.09 | 4.91 |
| | MoPs | 84.10 | 12.20 | 4.23 | 4.06 |
| | Gain ± | **+58.92 (70%)** | **+5.00 (41%)** | **+0.86 (20%)** | **+0.85 (21%)** |
| Natural Instruction | FedPrompt | 91.64 | 14.42 | 6.43 | 6.14 |
| | MoPs | 72.04 | 12.38 | 5.75 | 5.65 |
| | Gain ± | **+19.6 (27%)** | **+2.04 (16%)** | **+0.68 (12%)** | **+0.49 (9%)** |

Table 5: Summary of final perplexities reported on structured pruning in federated scenario,using a pool of 100 available clients, sampling 10 per iteration.

**Gating function analysis on the federated setup.** But, *why is MoP performing even better in FL settings?* Figure 3 illustrates that the pretraining step in federates provides useful information to the gating function to accurately capture the domain/task relationships. Tables 4 and 5 suggests that the gating function is beneficial in mitigating the model drift problem in the federated setting. This is because the gating function selectively updates the relevant experts related to each client, thus ensuring that the model updates are properly

aligned and preventing model drift. Consequently, the gating function plays a critical role in overcoming the heterogeneity of the data.

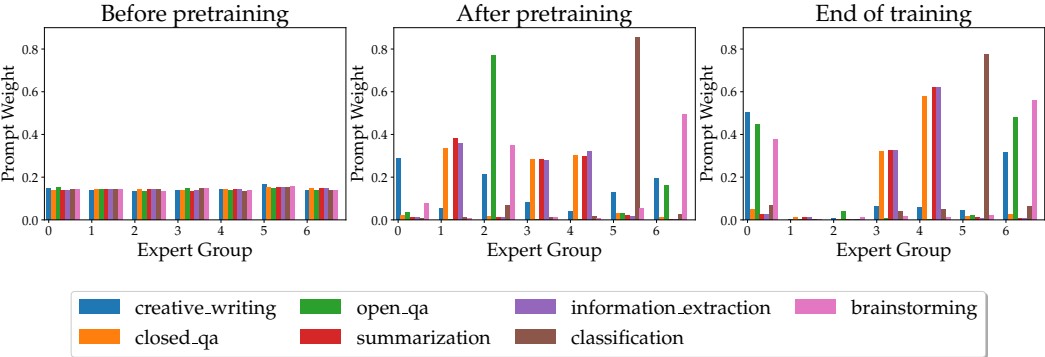

Figure 3: Averaged Prompt weight assigned each prompt group by gating function for test dataset using 3:4 (75%) structured pruning Llama-7B

**Quantization results.** FL is often limited by communication and computation constraints, so model compression methods such pruning and quantization are often used in combination. To test MoP, we combined `Int8` quantization with different pruning ratios in FL. As seen in Table 6 and Table 7, MoPs outperformed the baseline in all cases but two case. MoP achieved the best results with medium pruning ratio. This result suggests that the effectiveness of a gating network can be significantly impacted by the pruning ratio. If the pruning ratio is too aggressive, the gating network will be rendered ineffective due to the poor embedding network. On the other hand, if the pruning ratio is too low, there may not be enough room for improvement compared to the baseline.

| Dataset | Methods | Unstructured pruning (ratio) | | |
|---|---|---|---|---|
| | | `Int8`+90% | `Int8`+85% | `Int8`+75% |
| Dolly-15K | Baseline | 146.24 | 78.62 | 28.95 |
| | MoP | 140.05 | 71.25 | 28.26 |
| | Gain ± | **+6.19 (4%)** | **+7.37 (10%)** | **+0.69 (2%)** |

Table 6: `Int8` quantization with unstructured pruning results on Dolly-15 dataset in the federated learning scenario with 10 clients.

| Dataset | Methods | Structured pruning (ratio) | | | |
|---|---|---|---|---|---|
| | | `Int8`+7:8 (87.5%) | `Int8`+3:4 (75%) | `Int8`+2:4 (50%) | `Int8`+4:8 (50%) |
| Dolly-15K | Baseline | 192.10 | 50.30 | 14.24 | 13.13 |
| | MoP | 166.48 | 47.37 | 14.51 | 13.10 |
| | Gain ± | **+ 25.62(15%)** | **+2.93 (6%)** | -0.69 (2%) | +0.03 (0%) |

Table 7: `Int8` quantization with structured pruning results on Dolly-15 dataset in the federated learning scenario with 10 clients.

## 5 CONCLUSIONS

Our proposed gating function is able to identify relevant skills for the current task and dynamically select and combine prompts accordingly. This overcomes prompt training interference from multi-tasks across centralized and federated learning scenarios. Additionally, the results suggest that the gating function helps to overcome model drift problems resulting from heterogeneous data distribution in multi-source (federated) learning scenarios. This is achieved by locally selecting and updating only the relevant prompts for local data, which avoids training interference between clients. With no additional cost, the MoP method provides a powerful tool for overcoming interference from recovery of different skills from model compression, by embedding such skills in separated prompts. Overall, the MoP method is a promising approach for improving the efficiency and effectiveness of prompt-based learning systems.

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
