# Sweeping Heterogeneity with Smart MoPs: Mixture of Prompts for LLM Task Adaptation

## 1 Appendix

### A Federated skew distribution

To simulate a highly skewed data distribution in the across the clients for the federated learning experiments, we randomly selected total 5000 samples from all task categories. To simulate task and data heterogeneity, for data from each task category, we further split them into N partitions with different number of data samples (where N is the number of clients). To simulate the extreme data heterogeneity in real life scenario, we make one of the partition to have most of the data (it contains 15 times more samples than the rest partitions). We then randomly assigned one partition from each category to each client, resulting in different proportions and sizes of mixed tasks across the clients.

### B Centralized Training - Gating function Analysis

Below, we present the complete results of the averaged prompt weights assigned to each prompt group by the gating function before, during, and after training steps for the Dolly-15k dataset in the centralized setup. Different pruning ratios are displayed to demonstrate that more aggressive pruning ratios provide greater potential for improvement using the MoP method.

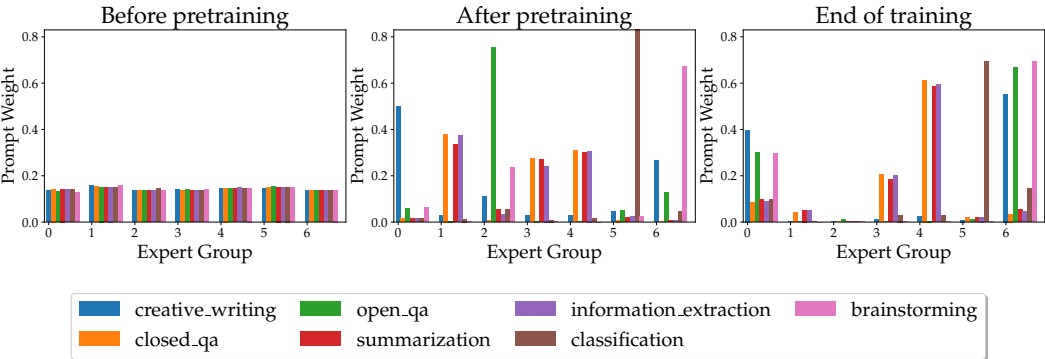

Figure 1: Averaged Prompt weight assigned each prompt group by gating function for test dataset using 75% unstructured pruning Llama-7B

### C Federated Training - Gating function Analysis

Similarly to the previous section, we show additional advantages provided by our method in the federated scenario. The alignment of the updates on the different experts helps minimize the effect of task interference.

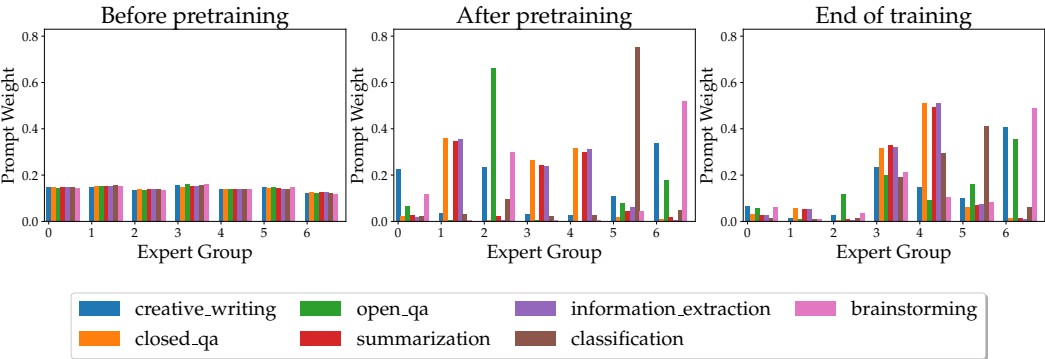

Figure 2: Averaged Prompt weight assigned each prompt group by gating function for test dataset using 90% unstructured pruning Llama-7B

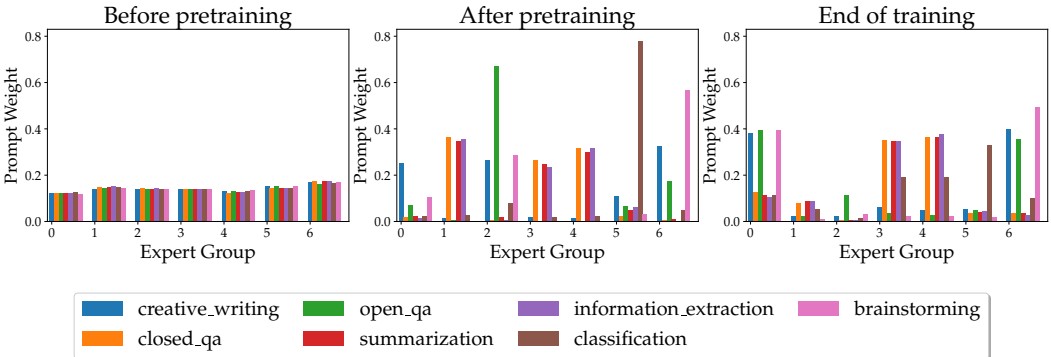

Figure 3: Averaged Prompt weight assigned each prompt group by gating function for test dataset using 7:8 (50%) structured pruning Llama-7B

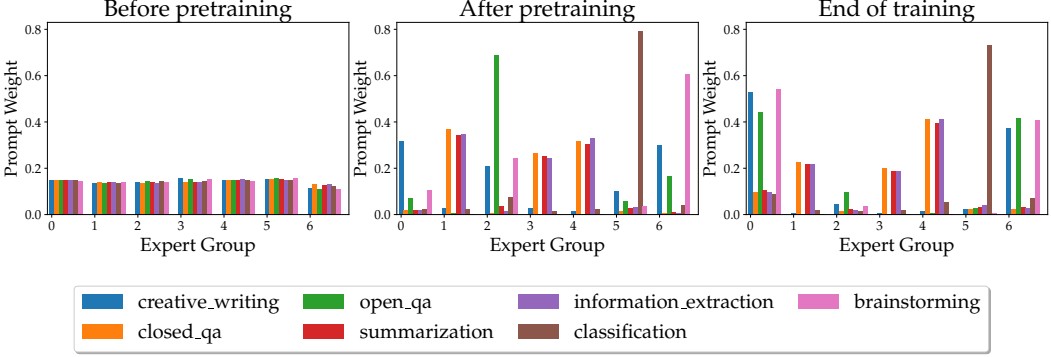

Figure 4: Averaged Prompt weight assigned each prompt group by gating function for test dataset using 3:4 (75%) structured pruning Llama-7B

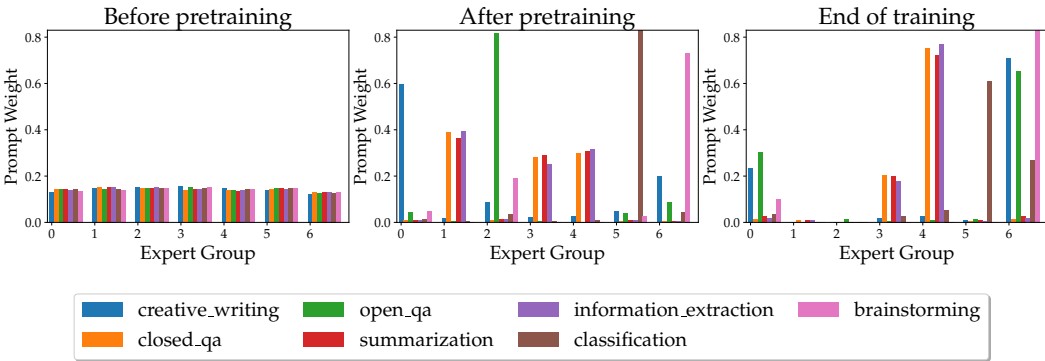

Figure 5: Averaged Prompt weight assigned each prompt group by gating function for test dataset using 2:4 (50%) structured pruning Llama-7B

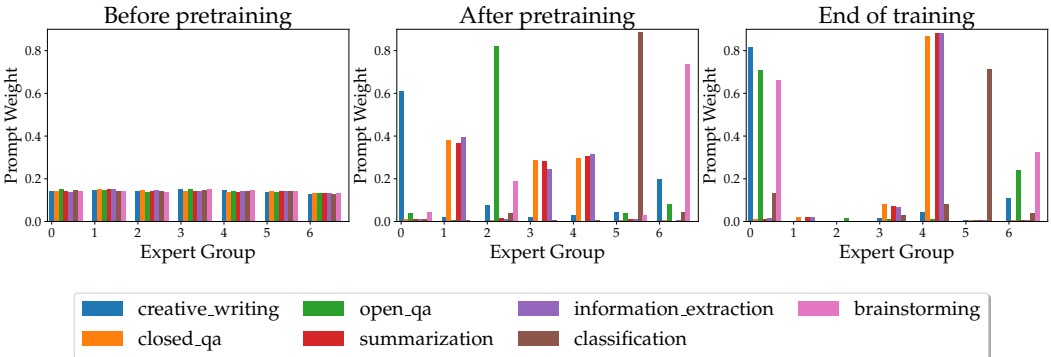

Figure 6: Averaged Prompt weight assigned each prompt group by gating function for test dataset using 4:8 (50%) structured pruning Llama-7B

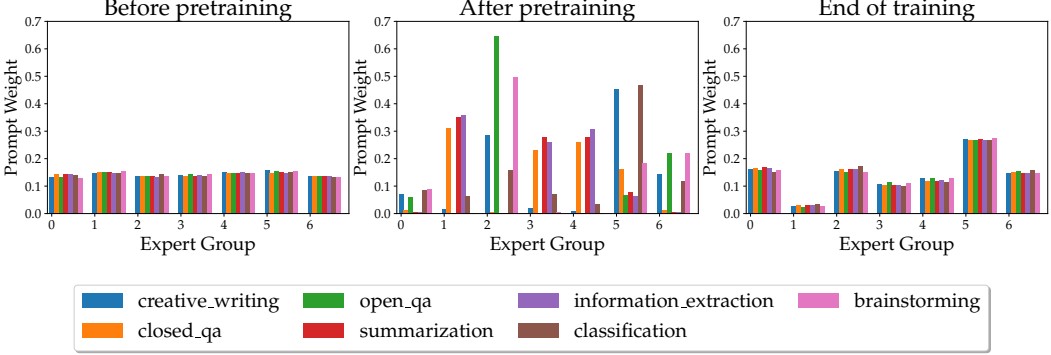

Figure 7: Averaged Prompt weight assigned each prompt group by gating function for test dataset using 75% unstructured pruning Llama-7B

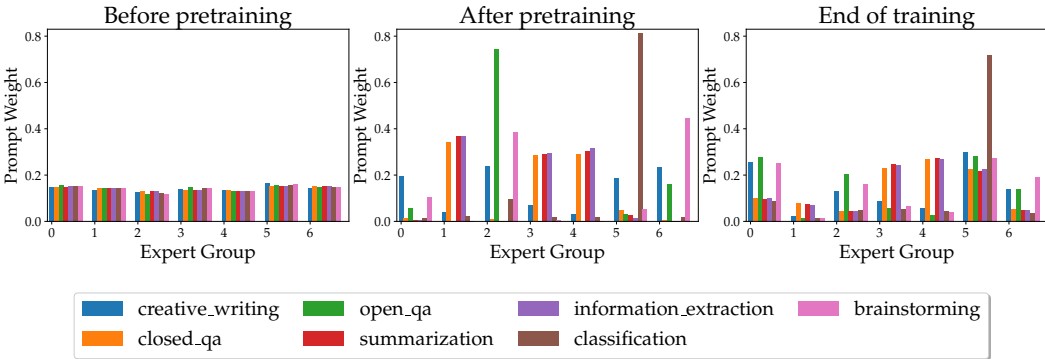

Figure 8: Averaged Prompt weight assigned each prompt group by gating function for test dataset using 85% unstructured pruning Llama-7B

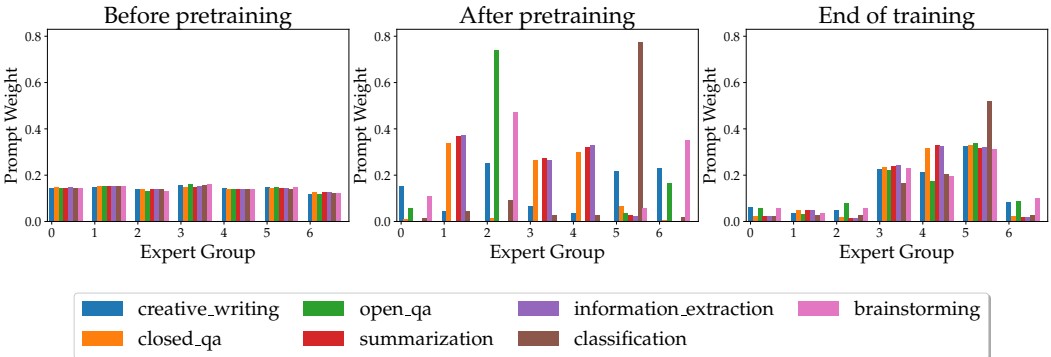

Figure 9: Averaged Prompt weight assigned each prompt group by gating function for test dataset using 90% unstructured pruning Llama-7B

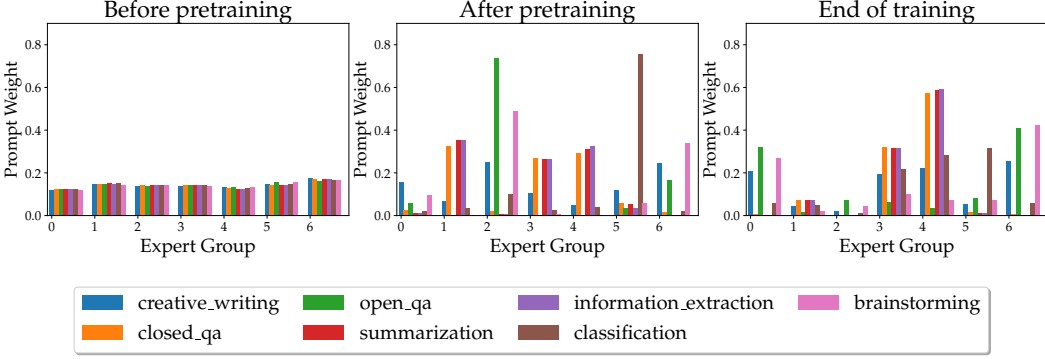

Figure 10: Averaged Prompt weight assigned each prompt group by gating function for test dataset using 7:8 (50%) structured pruning Llama-7B

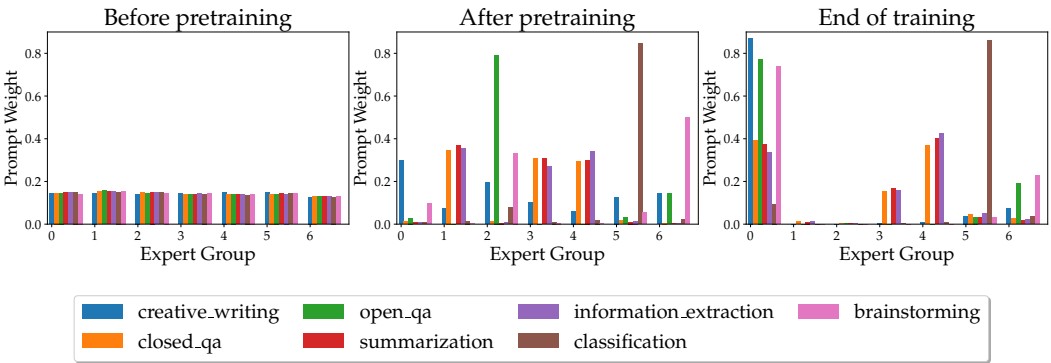

Figure 11: Averaged Prompt weight assigned each prompt group by gating function for test dataset using 2:4 (50%) structured pruning Llama-7B

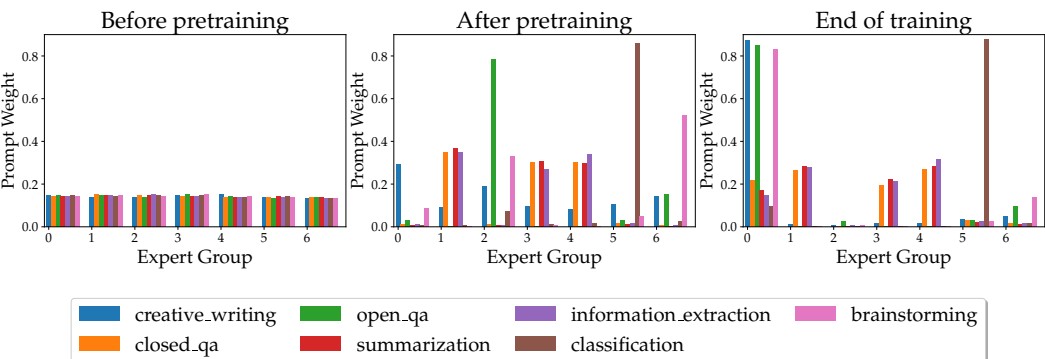

Figure 12: Averaged Prompt weight assigned each prompt group by gating function for test dataset using 4:8 (50%) structured pruning Llama-7B