# OpenReview forum: "Sweeping Heterogeneity with Smart MoPs: Mixture of Prompts for LLM Task Adaptation"
_ICLR.cc/2024/Conference — Submitted to ICLR 2024_

### Official Review · Reviewer_t2HG · 2023-10-24

**Soundness:** 3 good
**Presentation:** 2 fair
**Contribution:** 2 fair
**Rating:** 5
**Confidence:** 3

**Summary:**

This paper introduces a "mixture of prompts" technique, which selects relevant prompts for a prediction task based on a pre-trained routing module. In experiments, they find their method outperforms a prompt tuning baseline in a standard and federated learning setup interms of perplexity on Dolly and Super natural instructions.

**Strengths:**

- The idea of routing relevant prompts is useful -- it could make learning a new task quite rapid and only requirese inputting new prompts
- The presented results are promising in terms of improving performance. I think there are benefits of this method more generally in terms of ease of use -- it seems simpler to use than a prompt finetuning strategy.

**Weaknesses:**

There are some discussion points / experimental evaluations that would be useful to consider:
- A conceivable alternative to this method is including all available prompts in the context window of the language model for few-shot learning -- what's the motivation for using something like this method over this? Does it mitigate the "interference" described in the paper? Are there additional memory + computational constraints?
- Moreover, there are many peft techniques that have shown promise in learning from minimal labeled examples. I understand it may not be possible to baseline all of them, but including the most frequently used techniques, like LORA, would make the results more compelling to many readers to consider this method. I understand there are efficiency motivations here as well, but it would be useful for many readers in the centralization training I believe.
- There are a few simple baselines to consider for this method. In general, it seems like the method upweights relevant prompts in the weighting module for use for the task. It's fairly common just to select prompts for a particular task using KNN on embeddings from a larger set. How would this compare to the current technique? Or, is it expected to be worse for some reason?

**Questions:**

- Is this method sensitive to the ordering of prompts?
- Why perplexity for supernatural instructions? Dolly makes sense, but I believe this benchmark typically uses rouge-l -- I didn't see this choice explained and am curious.

---

> ### Author Response · Authors · 2023-11-21
> **Response to Reviwer t2HG**
>
> We thank the reviewer for the detailed and thoughtful review. Below, we respond to the points raised one by one. We hope our responses will resolve any further concerns, and we are available for other questions.
>
> -**Regarding including all available prompts**:
> Our baseline from [Xu 2023] indeed includes all available prompts in both training and inference. Our experiments and analysis show that when all prompts are used for diverse tasks (and cross-clients in federated learning), the training interference between different tasks and clients results in significant performance degradation, mitigated by our proposed gating function and expert prompts. %Our gating function is
>
> -**Regarding comparison with other PEFT methods**: We acknowledge several PEFT methods in current practice. We argue that prompt tuning is more widely adopted for instruction tuning on LLMs due to its straightforward implementation, which does not require any modifications to the LLM model. Moreover, prompt tuning is more parameter efficient compared to the adapter family. Therefore, to be as efficient and realistic as possible in our paper, we focus exclusively on the prompt tuning framework; however, we acknowledge that such experimental comparison is interesting.
> Finally, prompt tuning is more suitable for our gating function as it is more natural to select combinations of subsets of prompts depending on the current input rather than combining different adapters.
>
> -**Regarding other designs of a gating function**: We thank the reviewer for the exciting idea of different designs of the gating function.
> Directly using kNN to select from prompts still needs embedding the current input question from another embedding network, which might result in more computation and memory cost.
> In our method, the embedding for our gating function is directly reused from the intermediate layer embedding from the original LLM.
> Further, as all our prompts are randomly initialized during training, our gating function additionally needs to discover the set of skills across diverse tasks and guide the specialization of expert prompts towards such skills.
> This can only be done using a trainable gating network instead of a fixed clustering scheme, as in kNN.
>
> -**Regarding the ordering of prompts**: As our prompts are randomly initialized, our method does not assume any ordering of prompts before training. During training, we treat prompts as additional tokens with different position encoding.
>
> -**Regarding evaluation metrics**: Thank you for suggesting other metrics. We understand the importance of a unified evaluation metric across different tasks and datasets. That is why we chose to use PPL for our selected set of tasks, as it is particularly suitable for long text generation. However, we are open to incorporating more metrics in future drafts, and we appreciate your input.

---

### Official Review · Reviewer_nzzG · 2023-10-29

**Soundness:** 3 good
**Presentation:** 3 good
**Contribution:** 3 good
**Rating:** 5
**Confidence:** 3

**Summary:**

This paper proposes a mixture-of-prompts (MoP) method for multi-task, multi-source prompting tuning in LLMs. The key innovation of the method is that instead of using a single prompt to tune all tasks, which the authors claim could lead to degraded performance due to interference between tasks, MoP prompt tuning trains multiple soft prompts and uses a dynamic, learnable gating functionality to scale the attention weights in a task-dependent manner among the different prompts. The authors show the effectiveness of their methods in compressed LLMs and both centralised and federated setups and show improvement over baseline prompt tuning methods in the setups.

**Strengths:**

- The paper studies a specific problem setup that is both of great importance and is currently rather less extensively studied. While prompt tuning in different flavours has been extensively studied in the literature, multi-task, multi-source training in compressed and/or federated setup is not. Such a setup is both more challenging and more practical and thus is of great research and practical value to the community (some caveats to this point are stated in “Weaknesses”).
- The research problem is of relevance, the hypothesis is reasonable, and the method proposed is sound. Inspired by the mixture of expert literature, addressing the prompt tuning problem with heterogeneity with MoP is natural and intuitive. The benefits of the key ideas, such as the use of smart gating and its pretraining, are evidenced well by the experiments (such as Fig 2)

**Weaknesses:**

- While I appreciate the investigations in the setup as mentioned in “Strengths”, it seems that the method proposed is generic and not catered specifically to the federated or compressed setup, and these considerations are orthogonal. The experiments, however, exclusively test the MoP method in models aggressively compressed with pruning. I’m not sure if this is due to computational constraints, but I wonder how the method will perform in a less aggressively compressed LLM. It seems that the extent of gain decreases as pruning moderates. The authors made a comment on Page 9 that “if the pruning ratio is too low, there may not be enough room for improvement.” — I think the paper would benefit if the authors could add more explanations as to why this is the case.
- I also have some concerns regarding baseline comparisons and discussions w.r.t. the literature. The authors only consider a baseline proposed by Xu et al., 2023, which is standard prompt tuning on a compressed model after compression takes place. However, several other, arguably stronger baselines addressing the similar problem of prompt interference in multitask prompt tuning setup, as identified in Sec 3, are available, like [Wang 2023] (who seems to show that using a single prompt for many tasks works fine as long as we do low-rank task-specific updates afterwards) and [Vu 2022] — these works do not specifically target the compressed setup, but there’s no reason why they can’t work in it; it would be nice if the authors could discuss these works qualitatively and quantitatively. Additionally, without a comparison, I feel that the authors claim that their methods synergise with compression/federation learning setups particularly well (*“Via experiments, we have observed an emerging ability of MoPs: they work out of the box, regardless of any reasonable model compression ratio or technique (i.e. pruning, quantization)”* is a bit controversial, as the baseline prompt tuning methods may potentially work *generally* very well (note that Xu et al., 2023 that the authors compared against already shows such a trend) and the good performance is not due to the MoP design *specifically*. If it is indeed the case, the argument should be toned down a bit, in my opinion.
    - Several works use the mixture of prompt/adapter ideas that are missing, such as [Wang 2022], although I agree that the motivation or the problem setup is not identical. In many of these works mentioned above and works like Attempt (Asai et al., 2022, which the authors cited but did not discuss in detail), mechanisms similar to the gating function that authors claimed as a key contribution were proposed and I encourage the authors to compare and contrast.
- Better design ablations should be done: for example, the authors proposed to use the LLM intermediate embedding instead of a separate embedding for the gating function, and they decided to use the embedding after the 10th layer. These design decisions seem a bit ad-hoc; it’d be nice if ablations were done to demonstrate the impact of different design aspects on the final performance.


### Minor:
- It is better to dedicate Sec 2 entirely to background and related work for clarity of writing, but the authors blended their own proposals, such as the injection of prompts in the middle layers in this section. Separating the two components would help with clarity and flow.


## References
[Wang 2023] Wang, Z., Panda, R., Karlinsky, L., Feris, R., Sun, H., & Kim, Y. Multitask prompt tuning enables parameter-efficient transfer learning. ICLR 2023.

[Vu 2022] Vu, T., Lester, B., Constant, N., Al-Rfou, R., & Cer, D. Spot: Better frozen model adaptation through soft prompt transfer. ACL 2022

[Wang 2022] Wang, Y., Agarwal, S., Mukherjee, S., Liu, X., Gao, J., Awadallah, A. H., & Gao, J. AdaMix: Mixture-of-adaptations for parameter-efficient model tuning. EMNLP 2022

**Questions:**

Please address my concerns in *Weaknesses*.

I will reconsider the rating upon a satisfactory response and reading the other reviews.

---

> ### Author Response · Authors · 2023-11-21
> **Response to Reviwer nzzG (Part 1)**
>
> We thank the reviewer for the detailed and thoughtful review. Below, we respond to the points raised, one by one. We hope our responses resolve further concerns and are available for other questions.
>
> -**Regarding uncompressed LLMs**: Our method can be applied as a generic method agnostic to federated/centralized and compression ratios. However, we are motivated to focus on aggressively compressed LLM, as in real-life scenarios, users mostly have very limited computation and memory resources for LLM deployment.
> Unfortunately, we cannot train full-size LLM models on our GPUs to cover other scenarios due to limited computation and memory resources.
> We agree that, although our method achieves a consistently significant advantage compared with baselines in all pruning ratios, the absolute gain decreases when the pruning ratio goes down.
> From our experiments, it appears that this is because a higher compression ratio will impose more implicit prompt training interference since trainable prompts are responsible for both recovering more model capacity loss –due to higher compression– and model adaptation for downstream tasks.
> Yet, without additional experiments on full-size LLMs, this cannot be safely assumed for all compression ratios, and we will certainly look into this as future work.
>
> -**Regarding additional baselines [Wang 2023] [Vu 2022] [Wang 2022]**:
> We thank the reviewer for the thoughtful comments on the baselines used in our paper. Before discussing each of these papers in detail below, we state that it is a critical point in our motivation to be as close as possible to real-life LLM instruction tuning scenarios; this leads us to consider a limited set of baselines to be applied to our scenario.
>
> -**Regarding [Wang 2023]**:
> We argue that our proposed method and targeted scenario are different from those in [Wang 2023]. Firstly, our paper assumes the task-type/domain labels and questions and ground truth answers during training are unavailable. This is closer to a real-life scenario as the instruction tuning data are highly mixed collections of tasks asked by the diverse users without providing any task/domain labels. Yet, [Wang 2023] assumes knowledge of task labels for both the source and target tasks when performing prompt training and prompt distillation. Second, [Wang 2023] requires labeled data from target tasks on test clients to perform prompt distillation, which might not be available in a real-life case. Instead, we focus on a more realistic zero-shot scenario, requiring our gating function to select correct prompt experts for the current test client data without any knowledge (zero-shot) of ground truth label samples or label distribution of such test client. Finally, our method can be applied to centralized and federated learning scenarios. In federated learning scenarios, our approach is shown to automatically mitigate model drift problems in federated learning without any additional cost. Such might not be the case for the proposed method in [Wang 2023].

---

> ### Author Response · Authors · 2023-11-21
> **Response to Reviewer nzzG (Part 2)**
>
> -**Regarding [Vu 2022]**:
> We argue that [Vu 2022], similar to [Wang 2023], focuses on different scenarios and cannot be directly applicable to the settings described in our paper. In [Vu 2022], authors also assume the availability of task/domain labels during prompt tuning and prompt transfer for test clients, which, as mentioned above, is considered a significant limitation. Further [Vu 2022] considers two approaches: generic and targeted Soft Prompt Transfer (SPOT). The generic SPOT case is essentially the same as our prompt tuning baseline that we compare against in the paper, where the prompts are directly used for test clients. For targeted SPOT, it is required first to perform prompt tuning on test clients and then use similarity scores of the resulting prompt tokens to select source prompts. However, this violates the zero-shot assumption we have for the test clients, in which we do not have any ground truth label for any prompt tuning; thus, such a baseline cannot be used in our ablation studies.
>
> -**Regarding [Wang 2022] and [Asai 2022]**:
> We thank the reviewer for referencing other works proposing a mixture of expert frameworks. As [Wang 2022] only considers single-task PEFT training, the functionality of their gating network and adapters is different from our gating function and expert prompts. In particular, our gating function needs to discover the set of skills across diverse tasks and guide the specialization of expert prompts toward such skills. During inference, our gating function, given an unseen test client, needs to analyze the input question and correctly select the combination of relevant skills/expert prompts. In summary, we require our expert prompts to be highly specialized in different skills, which is in direct contrast with [Wang 2022], which utilizes a consistency regularization loss to encourage the similarity between expert adapters. Accordingly, our gating function cannot be a random routing network as in [Wang 2022], as it needs to analyze the current task and dynamically select relevant specialized skills/expert prompts.
>
> For the case of [Asai 2022], as we have mentioned in the related work section, [Asai 2022] assumes task labels for source prompt tuning, which are not available in our scenarios. Therefore, the gating function in [Asai 2022] does not need to discover the skills and guides the training of specialized experts in highly mixed task distribution without any task/domain labels.
>
> -**Regarding ablation studies**:
> We appreciate the reviewer's insightful comments. Our method uses LLM intermediate embedding instead of a separate embedding for the gating function to avoid the additional cost of training and inference incurred by having a different embedding network (such as another LLM or transformer model) for the gating function. We ensure a similar training/inference cost with a prompt tuning baseline by directly reusing the intermediate layer embedding. At the same time, to further reduce the computation costs (for both our method and baseline), we strategically add prompts only to the middle layers of the model, thus avoiding backpropagation of the full model during training. Therefore,  We set $L_{mid} = 10$ for an LLama-7B model with $L = 20$. Such choice of $L_{mid} = 10$ is to balance two conflicting requirements: (1) we want to inject prompts as late as possible to reduce back propagation cost during training and increase depth/capacity for the embedding network of the gating function, and (2) prompts should be injected early to have more capacity in influencing the pretrained LLM network.
>
> -**Regarding paper structure**: We appreciate the feedback and will separate those sections to better present our proposed method for the camera-ready version.

---

> > ### Comment · Reviewer_nzzG · 2023-11-22
> >
> > I thank the authors for their detailed feedback, and I've read the other reviews. The response has addressed several points satisfactorily (paper structure, related work), and I commend the authors for their efforts. However, my remaining concerns are:
> >
> > - Uncompressed LLMs: I appreciate and sympathize with the authors' computational constraints. However, I still believe that since the algorithm is not specifically catered to a compressed setup, there are confounding factors that can only be resolved if one tries the algorithm on an uncompressed LLM. I agree with the authors this is an important future direction.
> >
> > - Related works: I understand the contributions better with the response. I encourage the authors to add these discussions when they have the opportunity to revise, and I felt that even when a baseline is not fully compatible in terms of setup, it might still be useful to compare empirically as it gives some idea of how much the perform headroom is from the proposed method to another method potentially under a more relaxed setup. This part also seems to be the largest point of contention from the other reviews, and I encourage the authors to address them fully.
> >
> > - Ablation studies: While I thank the authors for explaining their design choices in detail, I still think empirical ablation studies would strengthen their argument more concretely.
> >
> > Thus, while the response has clarified the concerns and improved the strength of the manuscript, I still think the paper might benefit from an additional round of reviewing, so I will stick to my original rating for now.

---

### Official Review · Reviewer_YeGh · 2023-10-31

**Soundness:** 3 good
**Presentation:** 3 good
**Contribution:** 3 good
**Rating:** 5
**Confidence:** 4

**Summary:**

This paper proposed gating function is able to identify relevant skills for the current task and dynamically select and combine prompts accordingly. The authors claim that the proposed method can overcome prompt training interference from multi-tasks across centralized and federated learning scenarios.

**Strengths:**

This paper utilized a mixture of prompts to deal with heterogeneous tasks and data distributions. The performance seems to improved a lot compared with the baseline.

**Weaknesses:**

1. A mixture of prompts has been proposed in the NLP field in 2021 [1].

[1]. Qin, Guanghui, and Jason Eisner. "Learning how to ask: Querying LMs with mixtures of soft prompts." arXiv preprint arXiv:2104.06599 (2021).

2. The presentation of this paper makes me confused. Is the "Mixture of Prompts for LLM Task Adaptation" designed specifically for federated learning? I mean whether the method is designed for multi-task, multi-source scenarios. The multitask prompt tuning method proposed in [2] also enables parameter-efficient transfer learning. Is the method in this paper an incremental setting based on [2].
[2]. Zhen Wang, "Multitask Prompt Tuning Enables Parameter-Efficient Transfer Learning", ICLR 2023.

3. How do you observe the "aggressively compressed LLMs"? It seems that the authors utilized SparseGPT to get the compressed LLM. If so, any contribution to this point?

4. I think this paper combined many existing technologies into a good technique report.

**Questions:**

see the weaknesses above.

---

> ### Author Response · Authors · 2023-11-21
> **Response to Reviewer YeGh**
>
> We thank the reviewer for the review and references. Below, we respond to the points raised one by one. We hope our responses will resolve further concerns and are available for questions.
>
> -**Regarding comparison to [1]**:
> We thank the reviewer for pointing us to this paper. We argue that our proposed method and targeted scenario differ from those described in [1]. In particular, first, we focus on LLMs' standard instruction tuning scenario, which will include diverse tasks such as open-QA, closed-QA, creative writing, etc. On the other hand, [1] only utilizes relational datasets, focusing on tuning the relation extraction/inference abilities of language models. Secondly, we assume the task-type/domain labels are unavailable with questions and ground truth answers (as in $(x,y)$ pairs during training). This is closer to real-life scenarios as the instruction tuning data are highly mixed collections of tasks asked by the diverse users without providing any task/domain labels. Yet, in [1], authors assume the knowledge of task/domain label (the relation type $r$) for their gating function throughout the training. Thus, our gating function is different in architecture and training dynamics from the one in [1], as our gating function needs to discover the relevant set of skills in diverse tasks without any task/domain labels and guides the training of specialized expert prompts. As shown in Figures 2 and 3, our gating function successfully learns to select similar skill prompts for similar tasks without knowledge of task labels or similarity between tasks. This functionality is not straightforward whether it is possible in [1].
>
> -**Regarding comparison to [2]**:
> We again appreciate the reviewer's pointed paper. We argue that our proposed method and targeted scenario differ from [2]. In particular, for similar reasons in comparison to [1], [2] assumes knowledge of task labels for both the source and target tasks when performing prompt training and prompt distillation. This is a significant limitation since the real-life collected instruction training data mostly lacks task labels. Second, [2] requires labeled data from target tasks on test clients to perform prompt distillation, which might not be available in real-life cases. In contrast, we ask our gating function to select the correct prompt experts for the current test client data without any knowledge (zero-shot) of ground truth label samples or label distribution of such test clients. We agree that our method, as pointed out by the reviewer, can be applied to centralized and federated learning scenarios, further demonstrating its flexibility and robustness under different setups.
>
> -**Regarding any contributions on compressed LLMs**:
> Our primary motivation for focusing on aggressively compressed LLMs is to simulate real-life scenarios where users typically have limited computation and memory resources for LLM deployment. As stated in Section 4, we directly utilize SparseGPT as our compression technique. Our contribution in this area is to demonstrate that prompt tuning is much more difficult with higher compression ratios, but our proposed mixture of prompts framework can alleviate this.

---

### Official Review · Reviewer_g7Lw · 2023-11-06

**Soundness:** 2 fair
**Presentation:** 3 good
**Contribution:** 2 fair
**Rating:** 5
**Confidence:** 3

**Summary:**

This paper proposes an innovative method for improving the efficiency and adaptability of large language models (LLMs) in handling diverse tasks and data distributions. This method, called Mixture of Prompts (MoPs), uses smart gating functionality to dynamically select and combine different groups of prompts, acting as 'experts,' based on the task at hand.

**Strengths:**

+ This paper introduces a novel approach to prompt tuning using Mixture of Prompts (MoPs) with a smart gating function. This is a significant contribution as it addresses the challenge of task and data heterogeneity, which is particularly relevant for real-world applications of LLMs.

+ The paper demonstrates that the MoPs method is resilient to various model compression techniques, indicating that it can maintain performance even when computational resources are limited or when efficiency is a priority.

+ The proposed method shows a substantial decrease in final perplexity compared to baselines in both federated and centralized scenarios. This empirical performance is well-documented through experiments, providing strong evidence for the effectiveness of MoPs.

**Weaknesses:**

- While the paper introduces an innovative approach, it does not thoroughly discuss the complexity involved in implementing the MoPs method. Practical application details are crucial for adoption, and this could be an area requiring further clarification.

- The experiments conducted to demonstrate the efficacy of the MoPs method are limited to only two datasets. A broader range of datasets and scenarios would provide a more comprehensive understanding of the method's performance and generalizability.

- Although the paper mentions the mitigation of model drift problems, there is a lack of in-depth analysis of how the MoPs approach ensures the model's stability over time, especially in federated learning scenarios where model drift is a significant concern.

**Questions:**

1. The gating function's ability to dynamically select and combine prompts is a key contribution. Could the authors elaborate on how this function can scale with increasingly complex tasks or larger sets of heterogeneous tasks?
2. While the method shows resilience to model compression, could there be a threshold of model complexity below which the MoPs method starts to underperform? Is there a way to anticipate or calculate this threshold?
3. Given that the paper showcases empirical results, is there a plan to evaluate the MoPs method in a live, real-world environment, where tasks are not predefined and data distribution can be unpredictable?

---

> ### Author Response · Authors · 2023-11-21
> **Response to Reviewer g7Lw (Part 1)**
>
> We thank the reviewer for the detailed and thoughtful review. Below, we respond to the points raised one by one. We hope our responses will resolve any further concerns, and we are available for any other questions.
>
> -**Regarding implementation details**:
> We thank the reviewer for the suggestion, and we will add more details on implementation in the future draft, which was shortened due to the page limit.
>
> Our proposed framework is simple (i.e., only adding the gating function as an additional module) and needs minimal modification from prompt tuning baseline and no hyper-parameter tuning. Our method can be divided into $i)$ the expert prompts and $ii)$ the gating function. For our expert prompts, we are using 70 prompts.
>
> To decrease the size/complexity of the gating function and increase the capacity of each expert, we further split prompts into seven groups, each with ten prompts: 10 prompts per task for a 7-task problem; but as the results we have shown, it is not necessary to know a priori the number of tasks, as the system adapts and utilizes the required amount of ``experts''; see, e.g., Figure 3 in the main text. Therefore, the gating function will generate expert weights for each group, and all prompts within each group are assigned the same expert weight from the gating function in the following layers.
>
> Our gating function is a simple shallow MLP network that uses intermediate layer embedding as input and output expert weight (a seven-dimensional vector) with softmax. We inject all expert prompts in the 10th layer (the middle layer in LLAMA-7B mode). As described in the paper, this further reduces the training cost by only needing backpropagating to the middle layer.
>
> -**Regarding the number of datasets used**: We understand your concerns about not using more datasets, but the computation resources available were a limitation for conducting more experiments. Nonetheless, we emphasize that the Dolly-15K and Super-Natural Instruction datasets cover many sub-tasks, are non-trivial/easy datasets, and are widely used as representative LLM instruction tuning datasets.
>
> -**Regarding analysis on mitigation of model drift**: First, we note that any analysis can only be restricted to an experimental study rather than theoretical research, given the task complexity and models involved. As shown in Figure 3, our gating function learns to assign separate sets of experts for different tasks and the same group for similar tasks throughout training. In federated learning scenarios, the gating function during local training will only update the relevant subset of experts based on the task type of local training data. During synchronization, we update each expert by only collecting its updates from those clients (with training data according to task type). Therefore, we avoid averaging expert models trained on different task types, mitigating the model drift problem. We provide more analysis on the gating function on different pruning ratios in our appendix sections B and C.
>
> -**Regarding the scalability of the gating function**:
> We thank the reviewer for this question. Regarding scalability concerning the number of tasks, as we have pointed out in Section 3.1, we intend our expert prompts not to be simple experts specialized in one task; in that case, we would trivialize our gating function to a simple task classification and only learn the one-to-one relationship between prompt and tasks. Instead, each expert prompt should represent expert knowledge of a particular skill. For each task, the gating function should learn to select/utilize a combination of relevant skills (prompts) for the current task. As shown in Figures 2 and 3, our gating function learns to choose a similar subset of prompts as skill experts for similar tasks. In this way, we decouple the number of tasks with the number of prompts or the size of the gating function, as we assume those tasks, although diverse, can all be solved with a combination of finite skills. This indicates (only experimentally) that our gating function and the collection of trained prompts will be robust with the number of tasks. Finally, we intentionally choose the Super-Instruct dataset to account for task complexity, as it contains more complex tasks than Dolly-15K. Our experimental results show that our method maintains its advantage in both datasets.

---

> ### Author Response · Authors · 2023-11-21
> **Response to Reviwer g7LW (Part 2)**
>
> -**Regarding how MoPs perform when lower complexity models are used**:
> Although our method achieves a significant PPL loss decrease in all pruning ratios compared to the baseline, we can observe a relatively significant degradation from 85\% to 90\% pruning ratio. Thus, we can assume such a threshold should be between these ratios. Of course, such "phase transitions" could happen at different levels if different families of models, tasks, and datasets are used; we indicate that such a phenomenon might happen, and we report that in this paper.
>
> -**Regarding evaluation on live, real-world datasets**:
> We acknowledge the value of testing on real-world datasets. However, we have focused on standard instruction tuning datasets due to the lack of access to such data and testing environments. This decision is based on two main reasons: (1) as these instruction tuning datasets contain task labels (which are not used during training), we can ensure the heterogeneity of task and data distribution in federated learning scenarios when we create local datasets; (2) we can also more easily visualize and analyze our gating function and expert prompts, as demonstrated in Figure 2 and 3.

---

### Author Response · Authors · 2023-11-21
**Overview of our rebuttal**

We would like to express our gratitude to the reviewers for their thorough comments. We agree that the general scenario can be improved to better emphasize the significance of our method, and we will ensure to incorporate this feedback into the camera-ready version. We have included more ablation studies regarding the gating function behavior in general, to make it simpler to comprehend the behavior during training, and we will expand on this further in the appendix of the camera version.

---

### Meta-Review · Area_Chair_wymL · 2023-12-06

**Metareview:**

This paper proposes a novel Mixture of Prompts (MoP) method for adapting large language models (LLMs) to diverse tasks and data distributions. MoPs dynamically select and combine different prompts based on the task, improving efficiency and adaptability. The paper cannot be accepted in its current form due to concerns about complexity, experimental validation, and offer deeper analysis of model drift mitigation.

**Justification For Why Not Higher Score:**

- Limited experimental scope: The paper only evaluates in limited settings, leaving questions about its generalizability to broader scenarios. Also more strong baselines would be required to improve the convincingness of the method.
- Complexity and practical application considerations: The paper lacks a thorough discussion on the complexity, which is crucial for adoption.
- Lack of in depth analysis: While the paper mentions mitigating model drift, there is a lack of in-depth analysis on how to ensure stability, especially in federated learning.
- Similar to some prior work, insufficient novelty

**Justification For Why Not Lower Score:**

- MoP successfully addresses task and data heterogeneity: The paper introduces a novel MoP approach with a smart gating function that dynamically selects and combines different prompts based on the task. This is particularly relevant for real-world applications of LLMs with diverse data.
- Resilient to model compression: MoPs maintain performance even when computational resources are limited or efficiency is a priority. This makes it valuable for practical deployment scenarios.
- Empirical performance improvements: MoPs show a substantial decrease in final perplexity compared to baselines in both federated and centralized settings. This provides strong evidence for its effectiveness.

---

### Decision · Program_Chairs · 2024-01-16

Reject